# Sequestration and activation of plant toxins protect the western corn rootworm from enemies at multiple trophic levels

Christelle AM Robert[1,2]*, Xi Zhang[1†], Ricardo AR Machado[1†], Stefanie Schirmer[2], Martina Lori[1‡], Pierre Mateo[3], Matthias Erb[1], Jonathan Gershenzon[2]

[1]Institute of Plant Sciences, University of Bern, Bern, Switzerland; [2]Department of Biochemistry, Max Planck Institute for Chemical Ecology, Jena, Germany; [3]Laboratory of Fundamental and Applied Research in Chemical Ecology, University of Neuchâtel, Neuchâtel, Switzerland

*For correspondence:
christelle.robert@ips.unibe.ch

†These authors contributed equally to this work

Present address: ‡Department for Soil Sciences, Research Institute for Organic Agriculture, Frick, Switzerland

Competing interests: The authors declare that no competing interests exist.

**Abstract** Highly adapted herbivores can phenocopy two-component systems by stabilizing, sequestering and reactivating plant toxins. However, whether these traits protect herbivores against their enemies is poorly understood. We demonstrate that the western corn rootworm *Diabrotica virgifera virgifera*, the most damaging maize pest on the planet, specifically accumulates the root-derived benzoxazinoid glucosides HDMBOA-Glc and MBOA-Glc. MBOA-Glc is produced by *D. virgifera* through stabilization of the benzoxazinoid breakdown product MBOA by N-glycosylation. The larvae can hydrolyze HDMBOA-Glc, but not MBOA-Glc, to produce toxic MBOA upon predator attack. Accumulation of benzoxazinoids renders *D. virgifera* highly resistant to nematodes which inject and feed on entomopathogenic symbiotic bacteria. While HDMBOA-Glc and MBOA reduce the growth and infectivity of both the nematodes and the bacteria, MBOA-Glc repels infective juvenile nematodes. Our results illustrate how herbivores combine stabilized and reactivated plant toxins to defend themselves against a deadly symbiosis between the third and the fourth trophic level enemies.

DOI: https://doi.org/10.7554/eLife.29307.001

## Introduction

The growth and reproduction of herbivores is constrained by both plant quality and predation by higher trophic levels (*Hunter et al., 1997*; *Ode, 2006*). Certain herbivores have found a way out of this quandary by redirecting plant defenses against their own predators: By ingesting and accumulating plant toxins, a phenomenon referred to as sequestration, herbivores may make themselves unattractive or toxic to natural enemies (*Nishida, 2002*; *Petschenka and Agrawal, 2016*). The resulting transfer of plant toxins across three trophic levels is increasingly recognized as a powerful force that shapes the distribution and abundance of plants, herbivores and predators in natural and agricultural ecosystems (*Kumar et al., 2014*; *Hopkins et al., 2009*).

Many plants store non-toxic forms of plant defenses (so called protoxins) separately from the enzymes that activate them and only form the toxins upon tissue disruption when protoxin and activating enzyme come together (*Hopkins et al., 2009*; *Wouters et al., 2016*). Some sequestering herbivores in turn have evolved the ability to phenocopy these two-component defense systems by stabilizing and sequestering the protoxins and producing their own activating enzymes to release the toxins in a controlled fashion (*Jones et al., 2001*; *Beran et al., 2014*; *Zagrobelny and Møller, 2011*; *Kazana et al., 2007*; *Francis et al., 2002*; *Opitz and Müller, 2009*; *Müller et al., 2001*;

**eLife digest** The western corn rootworm is the most damaging pest of maize plants. Out of sight, the larvae of this beetle feed on maize roots, and cause billions of dollars worth of losses each year. One of the reasons why this pest remains such a problem is it can adapt and resist many crop protection strategies.

Biological control refers to combating a pest using its own natural enemies – for example, its predators. Biological control of the western corn rootworm has been attempted using nematode worms. Normally, the nematodes locate and enter an insect larvae, release bacteria that kill it, and then feed and multiply within the dead larvae. Yet, the western corn rootworm seems at least partly able to resist these nematodes, and the success of biological control in the field has been variable.

Several insect herbivores are known to accumulate, or sequester, plant toxins in their own body for self-defense. Previously, in 2012, researchers reported that the western corn rootworm is resistant and attracted to the major toxins in maize roots, the benzoxazinoids. The blood-like fluid of the western corn rootworm also repels many predators. Could the western corn rootworm be sequestering maize benzoxazinoids to resist the biological control of nematodes and their bacterial partners?

Plants store benzoxazinoids in a non-toxic form. If herbivores damage the plant, these molecules quickly break down into compounds that are toxic to most insects. Now Robert et al. – who include two of the researchers involved in the 2012 study – show that the western corn rootworm uses a similar defense system to protect itself against biological control nematodes and their bacterial partners. First, the larvae convert a benzoxazinoid breakdown product by adding a glucose molecule. They then release large amounts of this modified molecule to repel young nematodes. Second, via an unknown mechanism, the larvae stabilize a second plant-derived benzoxazinoid, sequester its non-toxic form in their bodies, and activate it upon nematode attack. The resulting toxins can kill both nematodes and their bacterial partners. By combining different chemical strategies to stabilize and activate plant toxins, the western corn rootworm is able to resist the nematodes used for biological control.

These findings can help to explain why biological control has had limited success against the western corn rootworm. In the long run, they may lead to more effective biological control programs, for instance by stopping the western corn rootworm from sequestering benzoxazinoids or by using natural enemies that are resistant to the insect's toxins.

DOI: https://doi.org/10.7554/eLife.29307.002

*Pontoppidan et al., 2001*; *Jones et al., 2002*). Although much progress has been made in the identification of the molecular mechanisms involved in phenocopying such two-component defense systems (*Beran et al., 2014*; *Opitz and Müller, 2009*; *Bridges et al., 2002*; *Hartmann et al., 1997*), we know surprisingly little about their actual function in defense. It remains for instance unclear whether re-activation is required for herbivore protection and which predators are targeted (*Beran et al., 2014*; *Zagrobelny and Møller, 2011*; *Kazana et al., 2007*; *Bridges et al., 2002*; *Hartmann et al., 1997*; *Zagrobelny et al., 2007*; *Pratt et al., 2008*; *Beran et al., 2011*; *Agrawal et al., 2012*). Testing whether two-component systems indeed protect herbivores against predators is necessary to place them in an adequate ecological and evolutionary context.

After being taken up by herbivores, plant defense metabolites may not only influence herbivore predators and parasitoids (*Ode, 2006*), but also higher trophic levels (*van Nouhuys et al., 2012*; *Harvey et al., 2007*). Effects extending to four trophic levels may be particularly likely in systems where the third and the fourth trophic level are intimately linked. Entomopathogenic nematodes (EPNs) for instance feed on insect-killing symbiotic bacteria which they inject into their herbivore hosts (*Kaya and Gaugler, 1993*). So far, the specific effects of sequestered plant defenses on the fourth trophic level remain unknown.

In contrast to aboveground herbivores, very little is known about plant toxin sequestration and the prevalence of two-component defense systems in belowground herbivores. Root feeders are among the most important agricultural pests and can have significant impacts on the abundance and distribution of other species, including leaf feeders (*Blossey and Hunt-Joshi, 2003*). The two

milkweed beetle larvae *Tetraopes tetraophthalmus* and *T. texanus* were recently found to accumulate cardenolides from their host plants. Sequestered cardenolide concentrations did not correlate with resistance to EPNs (*Ali and Agrawal, 2017*). Furthermore, the larvae and adults of the spotted cucumber beetle *Diabrotica undecimpunctata* accumulate cucurbitacin triterpenes from their host plants (*Tallamy et al., 1998*). Cucurbitacins are passed on to the eggs and can protect the latter against pathogenic fungi (*Tallamy et al., 1998*). Whether cucurbitacins protect *D. undecimpunctata* larvae against predators (*Gould and Massey, 1984*) and EPNs (*Barbercheck et al., 1995*) is unclear.

In this study, we investigated how the western corn rootworm *Diabrotica virgifera virgifera* (*D. virgifera*) deals with the major defensive metabolites of maize. *Diabrotica virgifera* is among the most damaging pest insects on this planet. Its larvae develop exclusively on maize roots and cause over $2 billion worth of damage every year in the United States alone (*Gray et al., 2009*). Earlier studies found that the hemolymph of the larvae is repellent to a wide variety of predators (*Lundgren et al., 2009*; *Welch and Lundgren, 2014*), which may contribute to the limited success of biological control programs against *D. virgifera* (*Gray et al., 2009*). Our own work revealed that the larvae of the western corn rootworm are fully tolerant to benzoxazinoids (BXs) (*Robert et al., 2012a*), a dominant class of secondary metabolites in maize that provides broad spectrum resistance against a variety of other pests and diseases (*Wouters et al., 2016*). We also found that the larvae are attracted by BXs and use them to navigate the rhizosphere and locate the most nutritious maize roots (*Robert et al., 2012a*). Based on these observations, we hypothesized that *D. virgifera* may be able to sequester BXs, and that these compounds may be the elusive repellent factor that renders *D. virgifera* repellent to predators. BXs function as classical two-component defenses, with the pro-toxins being stored in glycosylated form in the vacuoles of maize cells, the activating β-glucosidases being present in the cytosol and BX hydrolysis occurring upon tissue disruption (*Jonczyk et al., 2008*). We therefore also focused on understanding if and how the western corn rootworm stabilizes and re-activates these compounds for self-defense. EPNs are among the major natural enemies of western corn rootworm larvae and have been proposed as promising biocontrol agents to control the pest (*Kurtz et al., 2007*). Because of the intricate relationship between EPNs and their symbiotic bacteria (*Kaya and Gaugler, 1993*), we studied the impact of BX accumulation on both organisms.

## Results and discussion

### Sequestration of maize benzoxazinoids by the western corn rootworm

Metabolite analysis revealed that maize-fed *D. virgifera* larvae accumulated significant amounts of BXs in their body (*Figure 1A*). The highest concentrations were found for the glucosides HDMBOA-Glc (>100 µg/g FM) and MBOA-Glc (>25 µg/g FM). In contrast to *D. virgifera*, two generalist root feeders of the same genus, *D. balteata* and *D. undecimpunctata*, accumulated lower amounts of BXs (*Figure 1A*). In North and Central America, maize is attacked by many different *Diabrotica* species (*Szalanski et al., 2000*). Yet, only *D. virgifera* is fully specialized on maize and causes substantial yield losses (*Chiang, 1973*). Our experiments suggest that this specialization might be associated with the selective accumulation of BXs.

BX screening of different larval tissues showed that *D. virgifera* accumulates HDMBOA-Glc and MBOA-Glc predominantly in the hemolymph (*Figure 1B*). We also detected HDMBOA-Glc and MBOA-Glc on the exoskeleton, with estimated release rates of 3–6 ng/hr and larvae (*Figure 1B*). MBOA-Glc was predominant in the frass, at an average concentration of 11.4 mg*g FM$^{-1}$. Interestingly, the concentration of sequestered HDMBOA-Glc and MBOA-Glc varied between experiments, suggesting a possible impact of environmental conditions and/or small differences in larval age on sequestration patterns. A comparison of BX concentrations in maize roots and *D. virgifera* larvae that fed on these roots from hatching until third instar revealed that most larval BX levels mirror plant BX levels, with the exception of HDMBOA-Glc and MBOA-Glc. Although the BX abundance in the larvae were reduced by approximately 95%, HDMBOA-Glc levels were reduced by only 50% and MBOA-Glc was exclusively found in *D. virgifera* (*Figure 1C*).

Feeding *D. virgifera* larvae on BX-free *igl bx1* double mutant plants (*Ahmad et al., 2011*) led to the complete absence of BXs in the larvae (*Figure 1—figure supplement 1*), including MBOA-Glc, demonstrating that BXs in *D. virgifera* are plant-derived. By contrast, feeding on *bx1* mutant plants which show a 90% reduction in BX levels (*Maag et al., 2016*) still resulted in a significant, albeit

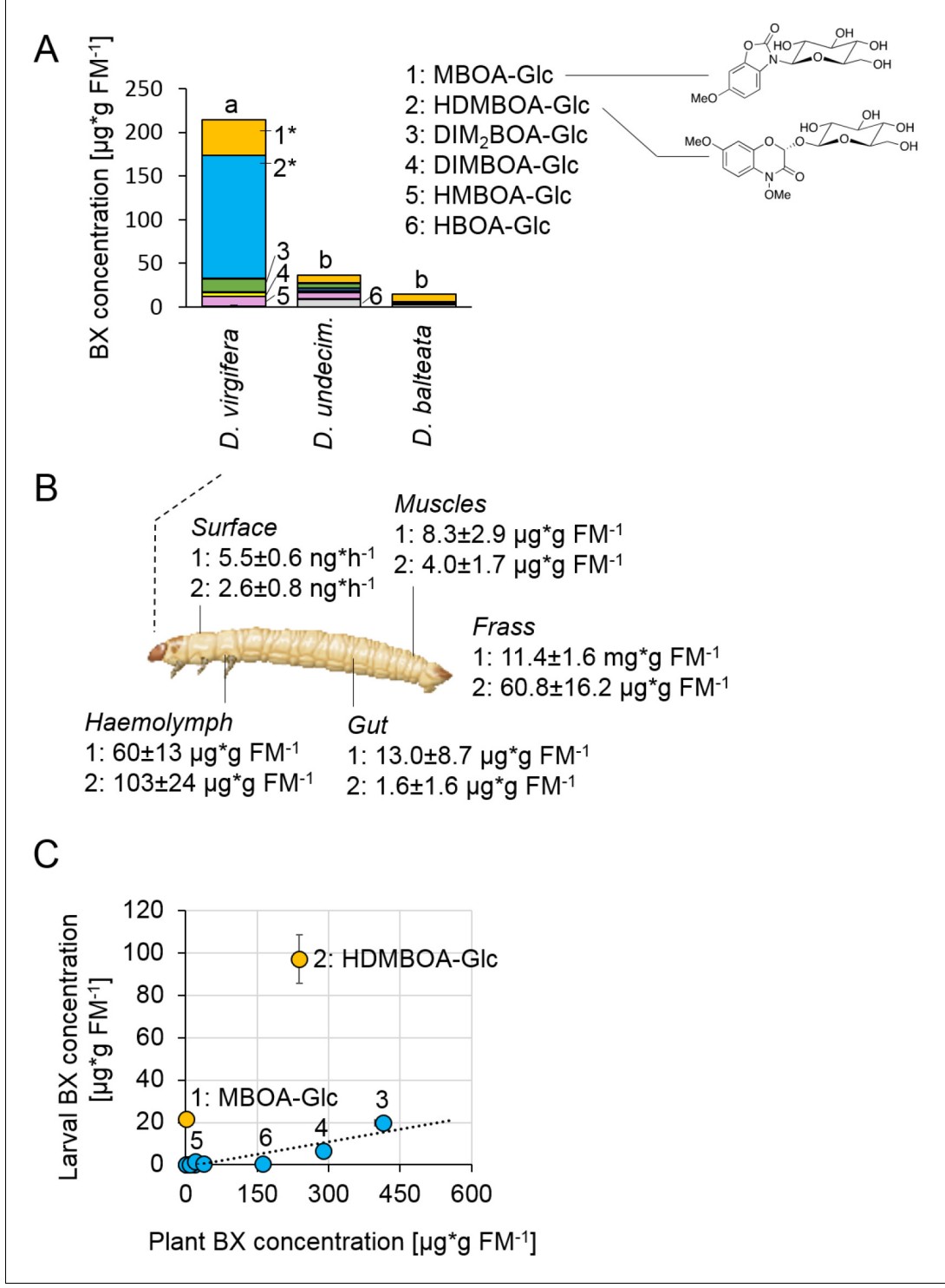

**Figure 1.** *Diabrotica virgifera* specifically and actively sequesters maize benzoxazinoids (BXs) (*Figure 1—figure supplements 1–3*). (A) BX concentration in larvae of the specialist *D. virgifera*, and the generalists *D. undecimpunctata* (*D.undecim.*) and *D. balteata*. Numbers denote the six most abundant BXs. Stars indicate significant differences between species (one-way ANOVA on transformed data (rank and square root transformations), $*p<0.05$). (B) BX concentrations in the haemolymph, gut, muscles, exudates (surface), and frass of *D. virgifera* larvae fed on wild-type B73 plants. (C) Correlation between BX concentrations in maize B73 plants and in third instar *D. virgifera* larvae that fed on those plants since hatching. Unlabeled blue dots correspond to other
*Figure 1 continued on next page*

*Figure 1 continued*

types of BXs. A linear regression between plant and larval concentrations is shown ($R^2$ = 0.8141, p=0.004, excl. MBOA-Glc and HDMBOA-Glc). Means ± SE are shown. Raw data are available in *Figure 1—source data 1*.
DOI: https://doi.org/10.7554/eLife.29307.003

The following source data and figure supplements are available for figure 1:

**Source data 1.** *Diabrotica virgifera* sequesters maize benzoxazinoids.
DOI: https://doi.org/10.7554/eLife.29307.007

**Figure supplement 1.** Benzoxazinoid levels in *Diabrotica virgifera* larvae fed on different maize lines.
DOI: https://doi.org/10.7554/eLife.29307.004

**Figure supplement 2.** Benzoxazinoid levels in *Diabrotica virgifera* larvae fed wild-type (B73) and *bx1* (bx1:B73) mutant plants.
DOI: https://doi.org/10.7554/eLife.29307.005

**Figure supplement 3.** Benzoxazinoid levels in aqueous surface extracts of *Diabrotica virgifera* larvae fed on wild-type (B73) and *bx1* (bx1:B73) mutant plants.
DOI: https://doi.org/10.7554/eLife.29307.006

lower accumulation of HDMBOA-Glc in *D. virgifera* larvae (*Figure 1—figure supplement 2*). HDMBOA-Glc release from the exoskeleton did not differ between wild type (WT) B73 and *bx1* mutant fed larvae, while MBOA-Glc release was significantly reduced (*Figure 1—figure supplement 3*). The observed accumulation patterns suggest a high degree of structural selectivity regarding uptake and sequestration of BXs by *D. virgifera*.

## Stabilization of benzoxazinoids in rootworm larvae and reactivation upon attack

We next investigated the processes which enable *D. virgifera* to sequester its two main BXs. MBOA, a benzoxazolinone-type BX, is a common product of DIMBOA and HDMBOA degradation in insect guts that has negative consequences for insect herbivores (*Wouters et al., 2016*; *Glauser et al., 2011*; *Maag et al., 2014*; *Wouters et al., 2014*). When *D. virgifera* gut extracts were incubated with MBOA, MBOA-Glc was readily formed (*Figure 2A*), suggesting that *D. virgifera* is capable of this N-glycosylation reaction as leaf feeding caterpillars are (*Maag et al., 2016*). Since MBOA-Glc was not deglycosylated by a root extract (*Figure 2—figure supplement 1*), it may represent a stable form of BX that can be absorbed from the gut by *D. virgifera* without toxic consequences.

In contrast to MBOA-Glc, HDMBOA-Glc is a common plant BX that is rapidly hydrolyzed by plant-derived β-glucosidases (*Glauser et al., 2011*), and the resulting aglycone is highly unstable (*Nishida, 1994*). To date, no other maize-feeding herbivore apart from *D. virgifera* is known to be able to accumulate HDMBOA-Glc (*Wouters et al., 2016*; *Glauser et al., 2011*; *Maag et al., 2014*; *Wouters et al., 2014*). To determine if *D. virgifera* larvae can inhibit HDMBOA-Glc hydrolysis, we incubated gut extracts with HDMBOA-Glc and β-glucosidase and analyzed aliquots of the extracts over time (5, 60 and 180 min) for BXs. HDMBOA-Glc was hydrolyzed both in presence and absence of the gut extracts (*Figure 2B*). To determine if *D. virgifera* can reglycosylate free HDMBOA, we incubated gut extracts with HDMBOA-Glc, β-glucosidase, and [$^{13}C_6$] UDP-glucose and analyzed aliquots of the extracts over time (5, 60, 180 min). No [$^{13}C$]-labeled HDMBOA-Glc was detected within the 3 hr. Thus, our *in vitro* experiments suggest that HDMBOA-Glc accumulation does not proceed via inhibition of HDMBOA-Glc hydrolysis in the *D. virgifera* gut nor by reglycosylation of free HDMBOA. Alternative strategies for HDMBOA-Glc accumulation in the hemolymph may include rapid transport (*Abdalsamee et al., 2014*) or transient stabilization through other chemical modifications (*Wang et al., 2012*).

Once sequestered, BXs can be of defensive value to herbivores if they can be reactivated. When we simulated predator attack by crushing the larvae with forceps, HDMBOA-Glc was rapidly broken down, and the final catabolite MBOA (*Maresh et al., 2006*) accumulated at concentrations of >25 μg/g FM (*Figure 2C*). No reduction in MBOA-Glc levels was observed (*Figure 2C*). When *D. virgifera* larvae were exposed to the EPN *Heterorhabditis bacteriophora*, MBOA concentrations increased as well, albeit at lower levels (*Figure 2D*). Together, these results show that *D. virgifera* produces MBOA via HDMBOA-Glc degradation upon predator and EPN attack. To understand whether BX glucosides may be activated by EPNs and their endobionts directly, we incubated them with purified

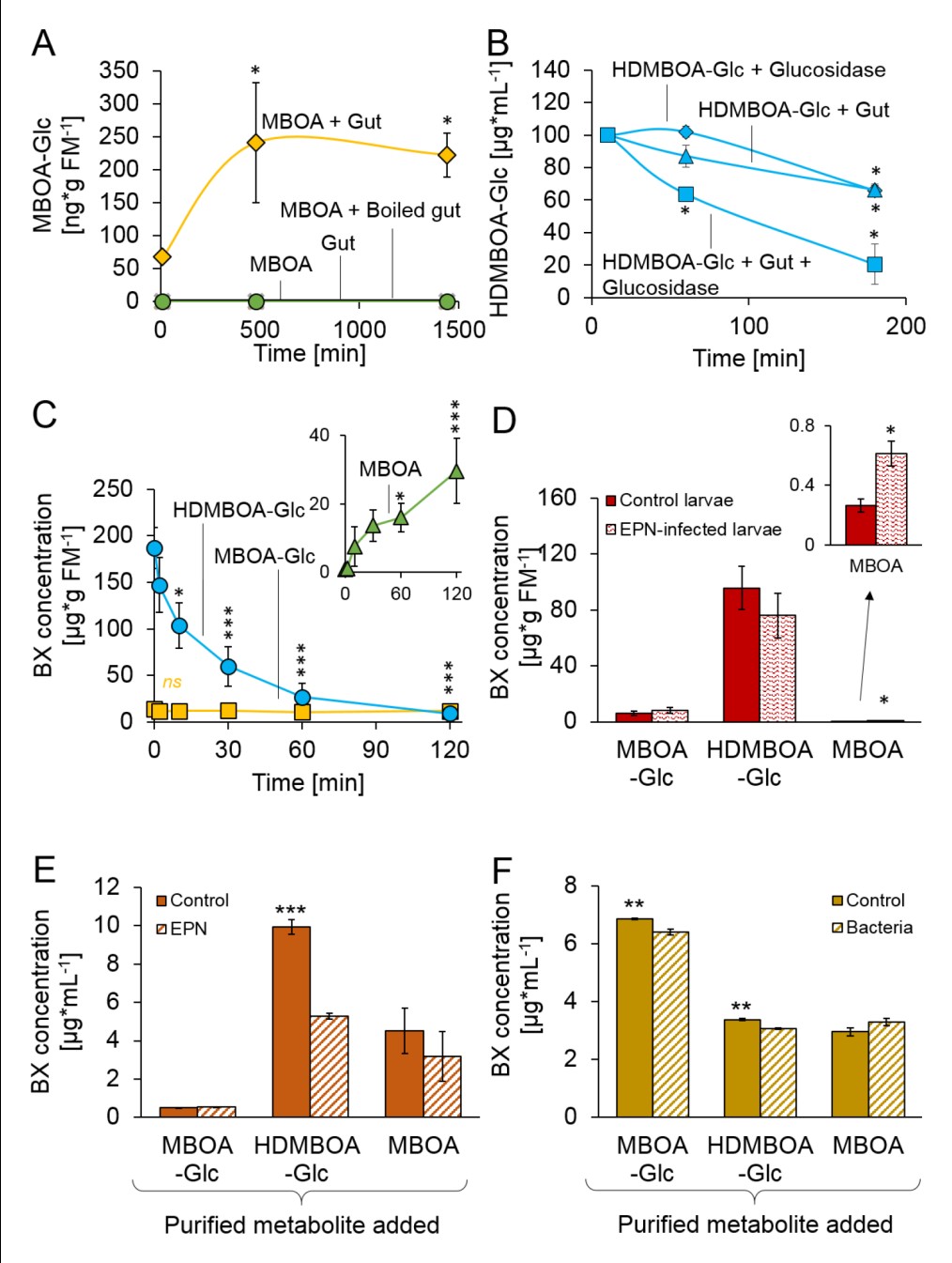

**Figure 2.** Stabilization and reactivation of stored benzoxazinoids (BXs) by *Diabrotica virgifera* and its natural enemies (*Figure 2—figure supplement 1*). (**A**) Stabilization of MBOA by conversion to MBOA-Glc in *D. virgifera* gut extracts. (**B**) HDMBOA-Glc deglucosylation in *D. virgifera* gut extracts. (**C**) BX reactivation in *D. virgifera* larvae upon mechanical tissue disruption. (**D**) BX reactivation in *D. virgifera* larvae upon exposure to the entomopathogenic nematode (EPN) *Heterorhabditis bacteriophora*. (**E**) BX reactivation by *H. bacteriophora* 24 hr after addition of purified metabolites. (**F**) BX reactivation by the EPN endosymbiotic bacterium *Photorhabdus luminescens* 24 hr after addition of purified metabolites. Means ± SE are shown. Stars indicate significant differences between time points (repeated measures ANOVAs, **A–C**) or between treatments (Student's t-tests, **D–F**; *p<0.05, **p<0.01, ***p<0.001). Raw data are available in *Figure 2—source data 1*.

DOI: https://doi.org/10.7554/eLife.29307.008

The following source data and figure supplement are available for figure 2:

**Source data 1.** *Diabrotica virgifera* stabilizes and reactivates stored benzoxazinoids.

*Figure 2 continued*

DOI: https://doi.org/10.7554/eLife.29307.010

**Figure supplement 1.** Degradation of MBOA-Glc by plant-derived hydrolases.

DOI: https://doi.org/10.7554/eLife.29307.009

HDMBOA-Glc, MBOA-Glc and MBOA. *Heterorhabditis bacteriophora* infective juveniles degraded HDMBOA-Glc to MBOA (*Figure 2E*), but not MBOA-Glc. MBOA was not further converted by *H. bacteriophora* (*Figure 2E*). The entomopathogenic bacterium, *Photorhabdus luminescens*, a symbiont of *H. bacteriophora*, catabolized both HDMBOA-Glc and MBOA-Glc, albeit at very low efficiency (*Figure 2F*). This work shows that the activation of sequestered plant toxins may occur upon contact with predator-derived factors, which may represent an additional route by which stabilized plant toxins can be used as anti-predator defenses by herbivores.

## Sequestered and reactivated benzoxazinoids provide resistance to predation

To investigate whether BX uptake increases *D. virgifera* resistance to EPNs, larvae that had been feeding on B73 wild type (WT) or *bx1* mutant plants were exposed to *H. bacteriophora* infective juveniles. Infectivity by EPNs was around 15% on WT-fed *D. virgifera*. On *bx1*-fed *D. virgifera*, infectivity increased to 40% (*Figure 3A*). Experiments with BX-deficient *bx1* and *bx2* mutants in the genetic background W22 (*Tzin et al., 2015*) confirmed that WT-fed *D. virgifera* are significantly more resistant to EPNs than larvae fed on BX-deficient mutants (*Figure 3A*). To further explore the potential of BXs to suppress EPN infectivity, we conducted a series of experiments with *H. bacteriophora* and its endobiont *P. luminescens* using pure BXs. At physiological doses, pre-exposure to HDMBOA-Glc suppressed *H. bacteriophora* infectivity toward the non-sequestering *D. balteata* by 50% (*Figure 3B*). MBOA-Glc and MBOA pre-exposure did not have any significant effect. HDMBOA-Glc and MBOA increased *H. bacteriophora* mortality *in vitro* by 10% and 20%, while MBOA-Glc had no significant effect (*Figure 3C*).

*Photorhabdus luminescens* growth was inhibited by MBOA starting at concentrations of 25 µg/g and HDMBOA-Glc starting at concentrations of 100 µg/g (*Figure 3D*, *Figure 3—figure supplement 1*). Again, MBOA-Glc did not have any effect. These experiments show that BX-dependent resistance of *D. virgifera* against EPNs is associated with strong toxicity of HDMBOA-Glc and MBOA against both the nematode and its endobiontic bacterium. Together with the mutant experiments, these data show that sequestered and reactivated BXs protect *D. virgifera* against predation by the third and the fourth trophic levels.

## Sequestered and stabilized benzoxazinoids repel predators

As BXs also accumulate on the exoskeleton of *D. virgifera* (*Figure 1B*), we hypothesized that they may interfere with EPN host location and preference. In choice tests, *H. bacteriophora* infective juveniles were significantly more attracted to *bx1*-fed *D. virgifera* larvae than larvae fed on WT plants (*Figure 4A*). The same pattern was found when aqueous surface extracts of *bx1*- and WT-fed larvae were compared (*Figure 4A*). Complementing surface extracts of BX-free *D. virgifera* with MBOA-Glc or a mixture of MBOA-Glc and HDMBOA-Glc at physiological doses significantly reduced their attractiveness for *H. bacteriophora* (*Figure 4B*). HDMBOA-Glc alone had no effect on EPN attraction. Thus, MBOA-Glc reduces the attractiveness of *D. virgifera* to EPNs.

Together, the results above show that *D. virgifera* stores BXs, which it stabilizes and re-activates to disrupt EPN infection at different levels (*Figure 5*). First, the relatively stable MBOA-Glc is released in the frass and on the exoskeleton as a repellent for host-searching infective juvenile nematodes. Second, HDMBOA-Glc is activated to produce MBOA, which reduces the growth of the symbiotic bacteria injected into the haemocoel by EPNs to kill and pre-digest the larvae. Third, HDMBOA-Glc and its reactivation products kill EPNs directly and thereby likely reduce the infectiveness of the next generation of emerging infective juveniles. By interfering with these different processes, *D. virgifera* larvae become highly resistant to EPNs. *D. virgifera* larvae are gregarious, and larvae from the same batch of eggs often feed together on the same host plant (*Robert et al.,*

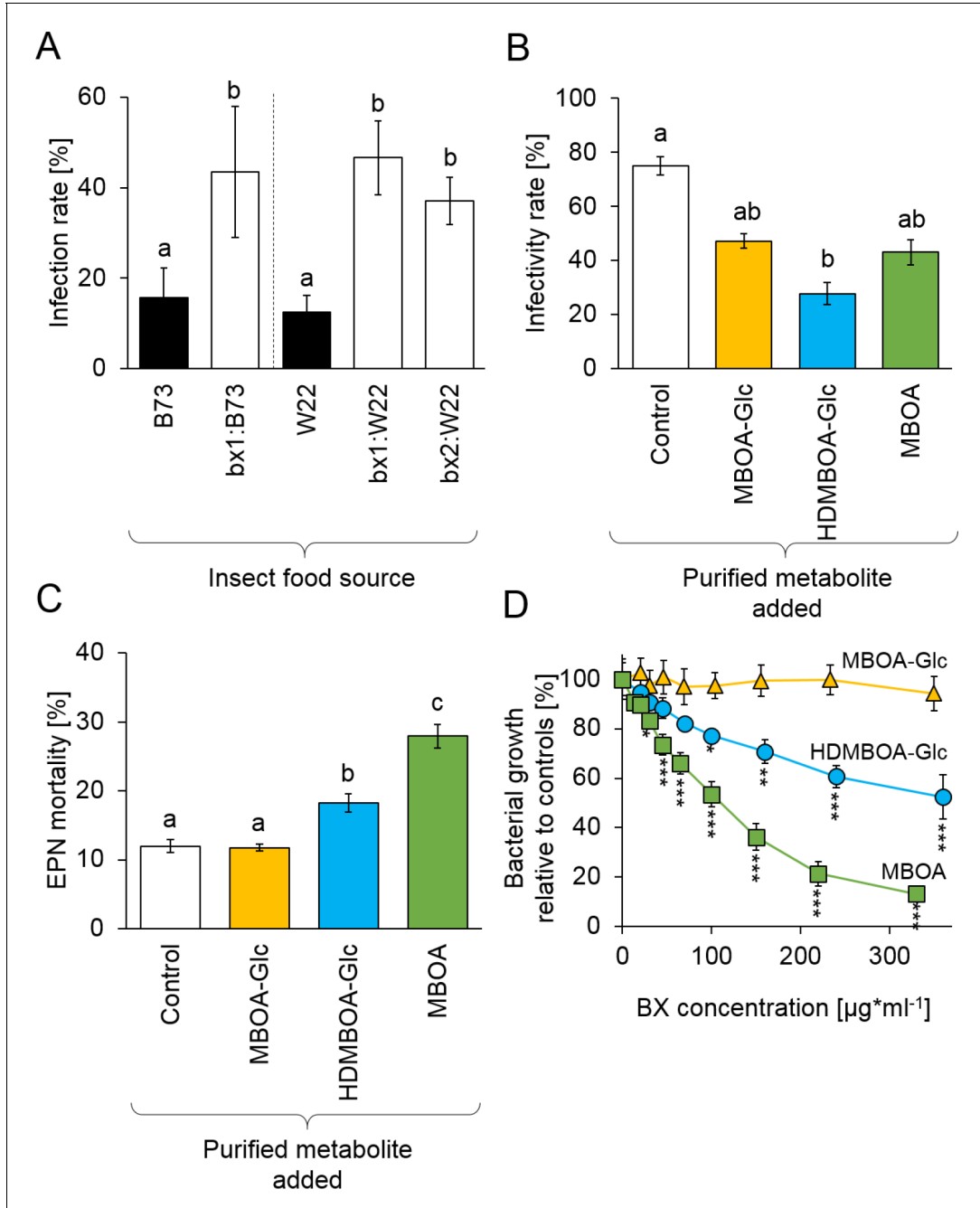

**Figure 3.** Benzoxazinoids (BXs) protect *Diabrotica virgifera* from its natural enemies (*Figure 3—figure supplement 1*). (**A**) Infection success by the entomopathogenic nematode (EPN) *Heterorhabditis bacteriophora* on *D. virgifera* larvae fed on WT (B73 and W22) or BX-deficient (bx1:B73, bx1:W22, bx2:W22) plants. (**B**) Effect of 7 days exposure to BXs on *H. bacteriophora* infectivity. (**C**) Effect of 7 days exposure to BXs on *H. bacteriophora* mortality. (**D**) Effect of BXs on the growth of the symbiotic entomopathogenic bacterium *Photorhabdus luminescens*. Different letters indicate significant differences between plant genotypes. Means ± SE are shown. Stars indicate significant differences between concentrations (A-C: one-way ANOVA, D: repeated measures ANOVA, *p<0.05, **p<0.01, ***p<0.001). Raw data are available in *Figure 3—source data 1*.

DOI: https://doi.org/10.7554/eLife.29307.011

The following source data and figure supplement are available for figure 3:

**Source data 1.** Benzoxazinoids protect *Diabrotica virgifera* from its natural enemies.

DOI: https://doi.org/10.7554/eLife.29307.013

**Figure supplement 1.** Growth curves and growth characteristics of *Photorhabdus luminescens* EN01 upon exposure to MBOA-Glc, HDMBOA-Glc and MBOA at different concentrations.

*Figure 3 continued on next page*

*Figure 3 continued*

DOI: https://doi.org/10.7554/eLife.29307.012

*2012b*). Reducing the build-up of high EPN densities within the rhizosphere may therefore also protect siblings from infection.

## Conclusions

The capacity to control the toxicity of plant secondary metabolites is crucial to the success of herbivores. Storing plant toxins for self-defense may be particularly advantageous, as it allows herbivores

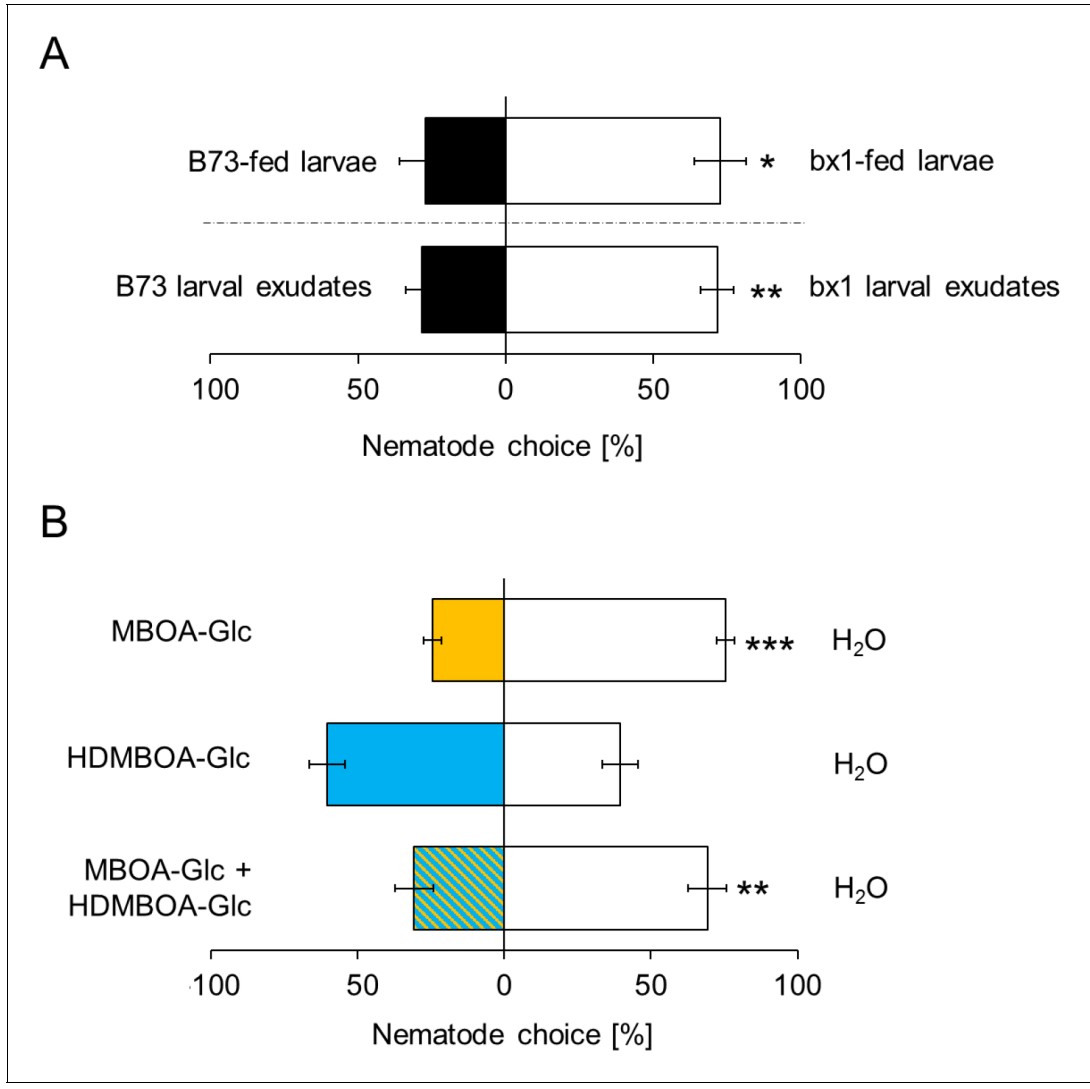

**Figure 4.** MBOA-Glc decreases the attractiveness of *Diabrotica virgifera* larvae. (**A**) Attraction of the entomopathogenic nematode (EPN) *Heterorhabditis bacteriophora to D. virgifera* larvae fed on wild-type (B73) and *bx1*-mutant (bx1:B73) (top) and aqueous surface extracts of larvae fed on wild type and *bx1*-mutant (bottom). (**B**) *H. bacteriophora* attraction to pure MBOA-Glc and HDMBOA-Glc at physiological concentrations. Means ± SE are shown. Letters indicate significant differences between treatments (one sample t-tests, *$p < 0.05$, **$p < 0.01$, ***$p < 0.001$). Raw data are available in *Figure 4—source data 1*.

DOI: https://doi.org/10.7554/eLife.29307.014

The following source data is available for figure 4:

**Source data 1.** MBOA-Glc decreases the attractiveness of *Diabrotica virgifera* larvae.

DOI: https://doi.org/10.7554/eLife.29307.015

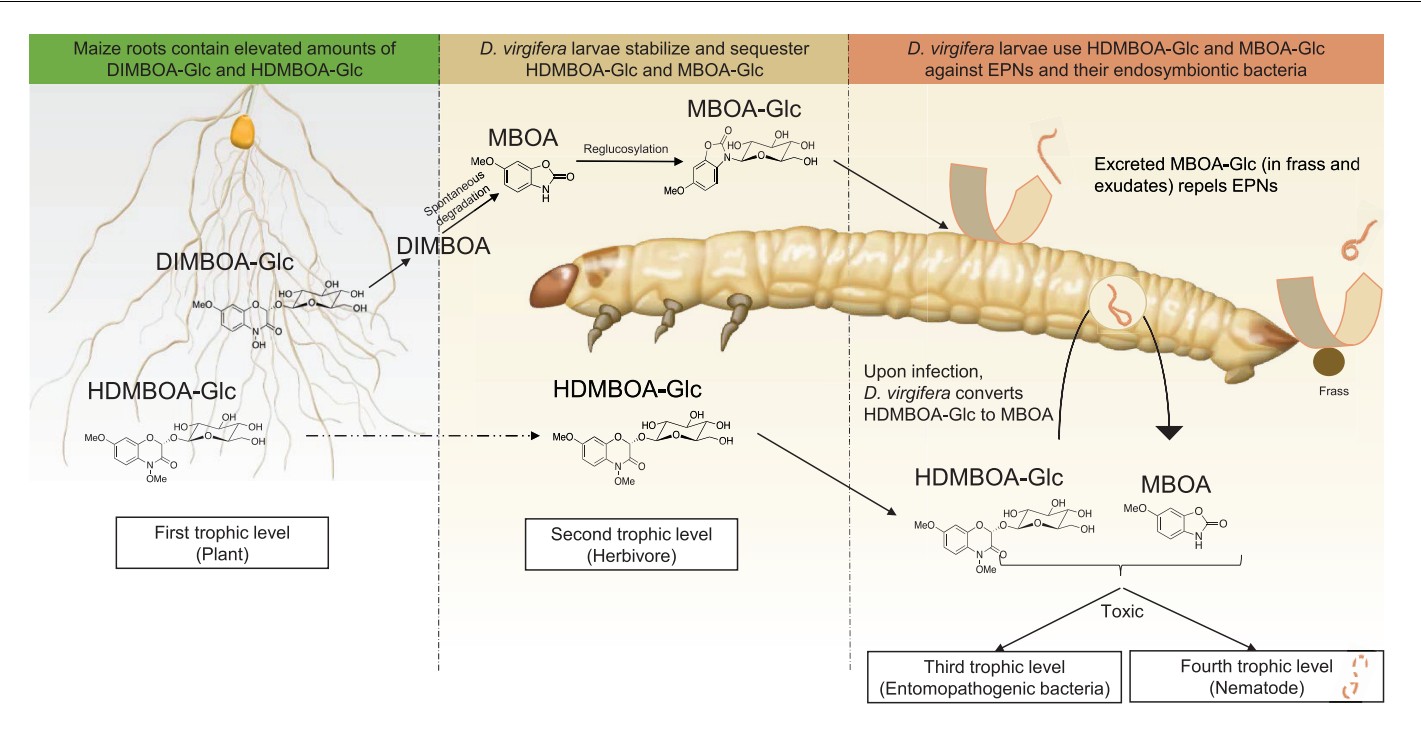

**Figure 5.** A model illustrating how BX sequestration and activation of plant toxins protects *Diabrotica virgifera* larvae from their enemies at multiple levels. MBOA-Glc, released in the frass and on the exoskeleton, repels infective juvenile entomopathogenic nematodes. Upon infection by nematodes and their symbiontic entomopathogenic bacteria, HDMBOA-Glc is activated to produce MBOA. Both HDMBOA-Glc and the activated MBOA reduce the growth of the symbiotic bacteria and kill EPNs.
DOI: https://doi.org/10.7554/eLife.29307.016

to escape bottom-up and top-down controls at the same time (*Kumar et al., 2014*; *Erb and Robert, 2016*). Our work illustrates how a specialized and highly destructive maize pest has evolved the ability to utilize the major toxins of its host plant to escape predation by soil-borne natural enemies. This protective effect is achieved through selective stabilization and reactivation of the toxins, which allows the herbivore to target different stages of the infection process of entomopathogenic nematodes and their endobiotic bacteria and thereby suppresses infection rates by over 50%. These results advance our basic understanding on the impact of plant chemistry on trophic cascades and provide an explanation for the limited success of biological control programs targeting the western corn rootworm (*Gray et al., 2009*).

## Materials and methods

### Biological resources and group allocation

Maize seeds (*Zea mays* L.) of the variety B73 were provided by Delley Semences et Plantes SA (Delley, CHE). The near-isogenic *bx1* mutant line in a B73 background was obtained by backcrossing the original *bx1* mutant five times into B73 (*Maag et al., 2016*). The wild-type W22, and the two *Ds* insertion mutant lines *bx1::Ds* (gene identifier GRMZM2G085381; Ds, B.W06.0775) and *bx2::Ds* (gene identifier GRMZM2G085661; Ds, I.S07.3472) (*Tzin et al., 2015*) were kindly provided by Georg Jander (Cornell University, Ithaca, NY; USA). The *igl.bx1* double mutant line 32R was obtained by crossing and backcrossing the two corresponding single mutant lines as described (*Ahmad et al., 2011*). Plants were grown in 120-mL plastic pots (Semadeni, Ostermundigen, CHE) filled with moist washed sand (1–4 mm, Landi Schweiz AG, Dotzigen, CHE) and a layer of 2 cm commercial soil (Selmaterra, Bigler Samen AG, Thun, CHE). Seedlings were grown in greenhouse conditions (23 ± 2°C, 60% relative humidity, 16:8 h L/D, and 250 µmol*m$^{-2}$*s$^{-1}$ additional light supplied by sodium

lamps). MioPlant Vegetable and Herbal Fertilizer (Migros, Zürich, Switzerland) was added every 2 days after plant emergence. Twelve-day-old plants were used for the experiments. *Diabrotica virgifera virgifera* (LeConte) eggs were generously supplied by USDA-ARS-NCARL, Brookings, SD. *Diabrotica balteata* (LeConte) eggs were kindly furnished by Syngenta (Syngenta Crop Protection AG, CHE). *Diabrotica undecimpunctata howardii* (Barber) eggs were bought from Crop Characteristics (Crop Characteristics Inc., Farmington, MN). Entomopathogenic nematodes *Heterorhabditis bacteriophora* were bought from Andermatt Biocontrol (Grossdietwil, CHE). The endosymbontic bacterium *Photorhabdus luminescens* EN01 was kindly provided by Carlos Molina (E-Nema Gesellschaft für Biotechnologie und Biologische Pflanzenschutz GmbH, Schwentinental, DE). All samples were randomly allocated to the different treatments. Whenever possible, data collection were made blindly.

## BX stabilization by the root herbivore

Stabilization of MBOA in *D. virgifera* guts was evaluated *in vitro* as follows: Third instar larval guts were collected and rinsed with distilled water. Five guts were pooled in 50 µL protein buffer containing 50 mM MOPSO (Acros, BE), 5 mM ascorbic acid (Sigma Aldrich Chemie GmbH, Schnelldorf, DE), 5 mM dithiothreitol (Sigma Aldrich Chemie GmbH), 10% (v/v) glycerol (Fluka Chemie GmbH, Buchs, CHE), 4% (w/v) polyvinylpyrrolidone (Sigma Aldrich Chemie GmbH), 0,1% (v/v) Tween 20 (Fluka Chemie GmbH) at pH 7 to which 0.25 mM MBOA (Sigma Aldrich Chemie) and 5 mM UDP-Glucose (Sigma Aldrich Chemie) were added (n = 3). Controls included (i) boiled guts (10 min at 100°C; n = 3), (ii) no gut (n = 3) or (iii) no MBOA (n = 3) in the buffer. Each pool of guts represented one biological replicate. All reactions were left at ambient temperature. After 10 min, 8 hr and 24 hr, 15 µL reaction solution was aliquoted and immediately mixed with 15 µL 100% MeOH (Fisher Scientific UK Ltd, Loughborough, UK). All extracts were vortexed and centrifuged at 14,000 rpm for 10 min at 4°C. Supernatants were collected for HPLC-MS analyses as described below.

The potential hydrolysis of HDMBOA-Glc was evaluated by adding purified HDMBOA-Glc to gut extracts. Gut extracts were prepared as described above. The protein buffer contained 3 µg/mL HDMBOA-Glc and 2 units of almond β-glucosidase (Sigma Aldrich Chemie) (n = 6). Controls included (i) HDMBOA-Glc and glucosidase in absence of gut (n = 11) and (ii) HDMBOA-Glc and gut only (n = 2). All reactions were left at ambient temperature. After 1 min, 1 hr and 3 hr, 15 µL reaction solution was aliquoted and immediately mixed with 15 µL 100% MeOH (Fisher Scientific UK Ltd). All extracts were vortexed and centrifuged at 14,000 rpm for 10 min at 4° C. Supernatants were collected for HPLC-MS analyses as described below. The proportion of initial HDMBOA-Glc remaining in the extract was calculated for each time point.

## BX reactivation by the root herbivore

MBOA-Glc and HDMBOA-Glc reactivation by *D. virgifera* larvae was evaluated in two experiments. Firstly, ten third instar *D. virgifera* larvae were collected and ground in the protein buffer described above (100 µL buffer per mg of collected larval tissue; n = 8). Aliquots were collected after 0, 2, 10, 30, 60 and 120 min and mixed with 100% MeOH (v/v). The resulting samples were vortexed, centrifuged at 14,000 rpm for 10 min at 4°C, and supernatants were used for HPLC-MS analyses. Secondly, BX reactivation was evaluated upon EPN infection *in vivo*. Fifty *D. virgifera* larvae were placed in a petri dish containing moist filter paper and 10,000 EPNs for 24 hr. Control larvae were placed in similar conditions, but without EPNs. *Diabrotica. virgifera* larvae were collected 40 hr after EPN exposure as a preliminary experiment had shown that the EPN endobiont starts growing at around that time. Three larvae of the same treatment were pooled together and ground in MeOH: H$_2$O: FA (50:50:0.5%; 100 µL buffer per mg of collected larval tissue; n $_{control}$=8, n $_{EPNexposed}$=13). The obtained extracts were vortexed and centrifuged at 14,000 rpm for 10 min at 4°C. Supernatants were collected for HPLC-MS analyses as described below.

## BX processing by EPNs

A thousand EPNs were placed in 1 mL tap water containing MBOA-Glc (2 µg/mL, n$_{control}$ = 10, n$_{MBOA}$ = 9), HDMBOA-Glc (20 µg/mL, n$_{control}$ = 6, n$_{HDMBOA-Glc}$ = 6) or MBOA (10 µg/mL, n$_{control}$ = 6, n$_{MBOA-Glc}$ = 6). Controls were in tap water only. After 24 hr, 1 mL of MeOH: FA (99:1%) was added to all samples. EPN samples were ground using a pellet pestle motor (Kimble Kontes, Sigma Aldrich

Chemie) for 30 s, vortexed and centrifuged at 14,000 rpm for 10 min at 4°C. Supernatants were collected for HPLC-MS analyses as described below.

## BX processing by bacteria

Standardized inoculums of growing *P. luminescens* bacteria were placed in 1 mL tap water containing MBOA-Glc (15 μg/mL, $n_{control}$ = 3, $n_{MBOA}$ = 6), HDMBOA-Glc (10 μg/mL, $n_{control}$ = 3, $n_{HDMBOA-Glc}$ = 6) or MBOA (10 μg/mL, $n_{control}$ = 3, $n_{MBOA-Glc}$ = 6). Controls were in tap water only. After 24 hr, 1 mL of MeOH: FA (99:1%) was added to all samples. resulting extracts were ground using a pellet pestle motor for 30 s, vortexed and centrifuged at 14,000 rpm for 10 min at 4°C. Supernatants were collected for HPLC-MS analyses as described below.

## Effects of BXs on EPN survival and infectivity

The effects of BX exposure on EPN survival were evaluated by incubating 1000 live EPNs in 1 mL of tap water containing either 50 μg MBOA-Glc, 150 μg HDMBOA-Glc or 25 μg MBOA (n = 10 per treatment, each biological replicate being the average of three technical replicates), and counting dead and living EPNs after 7 days. BX exposure effects on EPN infectivity were assessed by collecting living EPNs from the above solutions and placing them with non-sequestering *D. balteata* larvae. Briefly, a hundred EPNs in 500 μL tap water were added into petri dishes (9 cm diameter, Greiner Bio-One GmbH, Frickenhausen, DE) containing a filter paper and five third instar *D. balteata* larvae (n = 6 per treatment). All petri dishes were sealed with parafilm (Bemis Company Inc., Oshkosh, WI) to prevent the larvae from escaping. After 24 hr, *D. balteata* larvae were collected and placed into solo cups containing fresh crown root pieces. Five days later, all larvae were collected and dissected to determine their infection status. *In vivo* BX-mediated resistance to EPN infection was tested by exposing WT- and mutant fed *D. virgifera* larvae to EPNs in petri dishes as described above. The added roots after EPN exposure corresponded to the genotype the larvae had fed on prior to the experiment (mutant or WT). In a first experiment, *D. virgifera* larvae were grown on the *bx1* mutant (n = 9) and the near isogenic WT (B73; n = 8). In a second experiment, *D. virgifera* larvae were grown on the *bx1* (n = 9) and *bx2* (n = 7) *Ds* insertion mutants and their corresponding WT (W22; n = 8).

## Effects of BXs on bacterial growth

Inoculums of *P. luminescens* (initial optical density at 600 nm: $OD_{600}$ = 0.01) were grown in 70 μL of Luria Broth (LB) media (Carl Roth, Karlsruhe, DE) containing either water, MBOA, MBOA-Glc or HDMBOA-Glc at concentrations that ranged from 13 to 360 μg/mL (n = 4). Bacteria samples were incubated at 27 ± 0.02°C and analyzed over 30 hr using a Tecan Infinite M200 multimode microplate reader equipped with monochromator optics (Tecan Group Ltd., Männedorf, Switzerland). During incubation, the plate was shaken using orbital shaking (4.5 mm amplitude and 5 s shaking cycles) and the $OD_{660nm}$ was measured every 30 min. Data were analyzed using the Excel add-in DMfit (*Baranyi and Roberts, 1994*).

## EPN preference

To characterize the foraging behavior of EPNs, we designed a two choice assay in petri dishes. A 5 mm layer of 1% agar (Frontier Scientific Inc.) was poured in petri dishes (Greiner Bio-One). In a first experiment, one larva fed on B73 and one larva fed on the *bx1* mutant were pinned with a needle (Prym, DE) on each side of the dish (n = 10). Pinning the last segment of their abdomen did not kill the larvae but allowed them to move around the needle. In a second experiment, exudates of third instar *D. virgifera* larvae fed on B73 and on *bx1* mutant were collected by rinsing the larvae with 50 μL tap water of which 45 μL were added into two 5 mm diameter holes on each side of the petri dishes (n = 23). In a third experiment, the effect of BXs on EPN foraging behavior was tested by offering BX-complemented and control exudate extracts to the petri dishes. Exudates of larvae fed on the double *bx1-igl* mutant were collected as described above. HDMBOA-Glc (3.3 μg/mL; n = 23), MBOA-Glc (6.6 μg/mL; n = 18) or a mix of both (3.3 and 6.6 μg/mL HDMBOA-Glc and MBOA-Glc, respectively; n = 26) were added to 45 μL larval exudates. The final concentrations corresponded to natural concentrations found on third instar WT larval skin. Tap water was added to the exudates. A hundred *H. bacteriophora* in 50 μL tap water were added in a 5 mm diameter hole in the center of the agar plate. The number of EPNs in the four quarters of the plates were counted after 24 hr.

EPNs located in the quarters containing the treatment hole in their center were counted as choosing EPNs. Plates were no nematode moved from the center were excluded from the analysis.

## BX profiling

Plant samples were flash frozen and ground to a fine powder in liquid nitrogen. One milliliter extraction buffer (EB: MeOH: $H_2O$: formic acid (FA); 50: 50: 0.5%) was added to 100 mg sample. Larval samples were weighed (five larvae pooled per biological replicate) and directly ground in the extraction buffer (1 mL EB per 100 mg tissue) with a pellet pestle motor ($n_{D.balteata}$ = 3, $n_{D.undecimpunctata}$ = 8, $n_{D.virgifera}$ = 5). Larval exudates were extracted by rinsing third instar larvae with 25 μL distilled water and adding 25 μL MeOH: FA (99: 1%) to the solution (n = 6). Larval frass were collected by placing 20 starved larvae on maize roots for 2 hr before transferring them in 1.5 mL pre-weighed tubes for 1 hr (n = 7). BXs processed by EPNs (n = 6–9) and bacteria ($n_{control}$ = 3, $n_{bacteria}$ = 6) were extracted by adding MeOH: FA (99: 1%) to the EPN or bacteria solution (v/v) and ground using a pellet pestle motor for 30 s. All extracts were vortexed for 1 min and centrifuged at 14,000 rpm for 20 min, at 4°C. Supernatants were collected for HPLC-MS analyses. BX profiling of larvae of the three insect species and of *D. virgifera* larvae following disruption was conducted using an Agilent 1200 infinity system (Agilent Technologies, Santa Clara, CA) coupled to an API 3200 tandem spectrometer (Applied Biosystems, Darmstadt, DE) equipped with a Turbospray ion source following the method described elsewhere (*Handrick et al., 2016*). BXs in plants, infected larvae, larval tissue and exudates were quantified using an Acquity UHPLC system coupled to a G2-XS QTOF mass spectrometer equipped with an electrospray source (Waters). Gradient elution was performed on an Acquity BEH C18 column (2.1 × 50 mm i.d., 1.7 μm particle size) at 99–72.5% A over 3.5 min, 100% B over 2 min, holding at 99% A for 1 min, where A = 0.1% formic acid/water and B = 0.1% formic acid/acetonitrile. The flow rate was 0.4 mL/min. The temperature of the column was maintained at 40°C, and the injection volume was 1 μL. The QTOF MS was operated in negative mode. The data were acquired over an m/z range of 50–1200 with scans of 0.15 s at collision energy of 4 V and 0.2 s with a collision energy ramp from 10 to 40 V. The capillary and cone voltages were set to 2 kV and 20 V, respectively. The source temperature was maintained at 140°C, the desolvation was 400°C at 1000 L h-1 and cone gas flows was 50 L/hr. Accurate mass measurements (<2 ppm) were obtained by infusing a solution of leucin encephalin at 200 ng/mL at a flow rate of 10 μL/min through the Lock Spray probe (Waters).

BXs processed by EPNs and their endobiontic bacteria were analyzed with an Acquity UHPLC-MS system equipped with an electrospray source (Waters i-Class UHPLC-QDA, USA). Compounds were separated on an Acquity BEH C18 column (2.1 × 100 mm i.d., 1.7 μm particle size). Water (0.1% FA) and acetonitrile (0.1% FA) were employed as mobile phases A and B. The elution profile was: 0–9.65 min, 97–83.6% A in B; 9.65–13 min, 100% B; 13.1–15 min 97% A in B. The mobile phase flow rate was 0.4 mL/min. The column temperature was maintained at 40°C, and the injection volume was 5 μL. The MS was operated in negative mode, and data were acquired in scan range (m/z 150–650) using a cone voltage of 10V. HDMBOA-Glc and DIMBOA-Glc were quantified in positive mode using single ion monitoring (SIM) at m/z 194 with cone voltage of 20V. All other MS parameters were left at their default values as suggested by the manufacturer. Absolute BX concentrations were determined using standard curves obtained from purified DIMBOA, MBOA, DIMBOA-Glc, HDMBOA-Glc and synthetized MBOA-Glc.

## Statistical analysis

Statistical analyses were performed using SigmaPlot 13. Data were first tested for the heteroscedasticity of error variance and normality using Brown-Forsythe and Shapiro-Wilk tests. Data that did not fulfill the above assumptions were transformed or rank-transformed. Student t-tests and analyses of variance (ANOVA) were performed to assess differences between treatments. Repeated measures over time were analyzed using repeated measures ANOVAs (RM-ANOVA). Preference data were analyzed by comparing the average difference of the proportions of EPNs choosing control and treatment sides to the null Hypothesis $H_0$ = 0 using a one tailed t-test. Details on the data transformation, statistical tests and their outcome are available in the Summary Statistics.

## Acknowledgements

We thank Michael Reichelt for analytical support, Mareike Schirmer, Virginia Hill, Romain Daveu, Carla Arce, Liyong Zhang, Baptiste Biet, Lena Kurz, Amélie Tourat and Estelle Chassin for their technical help. We are also grateful to the insect rearing team and to IPS gardeners for their support. We also thank Thomas Degen for the drawings. This work was supported by the Swiss National Science Foundation (grants no. 160786, 155781, 157884, 140196).

## Additional information

### Funding

| Funder | Grant reference number | Author |
|---|---|---|
| Schweizerischer Nationalfonds zur Förderung der Wissenschaftlichen Forschung | 140196 | Christelle AM Robert |
| Schweizerischer Nationalfonds zur Förderung der Wissenschaftlichen Forschung | 160786 | Christelle AM Robert Matthias Erb |
| Schweizerischer Nationalfonds zur Förderung der Wissenschaftlichen Forschung | 155781 | Matthias Erb |
| Schweizerischer Nationalfonds zur Förderung der Wissenschaftlichen Forschung | 157884 | Matthias Erb |

The funders had no role in study design, data collection and interpretation, or the decision to submit the work for publication.

### Author contributions

Christelle AM Robert, Conceptualization, Data curation, Formal analysis, Supervision, Funding acquisition, Validation, Investigation, Visualization, Methodology, Writing—original draft; Xi Zhang, Ricardo AR Machado, Data curation, Formal analysis, Investigation, Visualization, Methodology; Stefanie Schirmer, Data curation, Formal analysis; Martina Lori, Data curation, Formal analysis, Visualization; Pierre Mateo, Resources, Investigation; Matthias Erb, Resources, Formal analysis, Funding acquisition, Investigation, Methodology, Writing—original draft; Jonathan Gershenzon, Resources, Methodology

### Author ORCIDs

Christelle AM Robert http://orcid.org/0000-0003-3415-2371
Matthias Erb http://orcid.org/0000-0002-4446-9834

### Decision letter and Author response

Decision letter https://doi.org/10.7554/eLife.29307.020
Author response https://doi.org/10.7554/eLife.29307.021

## Additional files

### Supplementary files

• Supplementary file 1. List of BX abbreviations.
DOI: https://doi.org/10.7554/eLife.29307.017

• Supplementary file 2. Summary statistics.
DOI: https://doi.org/10.7554/eLife.29307.018

• Transparent reporting form
DOI: https://doi.org/10.7554/eLife.29307.019

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
