## [Decision Letter]

Thank you for submitting your article "Sequestration and activation of plant toxins protects the western corn rootworm from enemies at multiple trophic levels" for consideration by *eLife*. Your article has been reviewed by two peer reviewers, and the evaluation has been overseen by Patricia Wittkopp as the Senior Editor and Reviewing Editor. The reviewers have opted to remain anonymous.

The reviewers have discussed the reviews with one another and the Reviewing Editor has drafted this decision to help you prepare a revised submission.

Summary:

This paper describes how western corn rootworms accumulate specific root-derived benzoxazinoid-glucosides to produce MBOA upon predator attack. MBOA not only makes the corn rootworm more resistant to nematodes but also repels them. This work provides a nice example of how a two-component defense system of a herbivore, using the prototoxins from the plant, is used.

More specifically, Robert et al. show that *Diabrotica virgifera* sequesters benzoxazinoid glucosides from maize roots. The larvae glycosylate the reactive breakdown product MBOA and thus convert it into a stable, supposedly non-toxic storage form (MBOA-glucoside). In addition, HDMBOA-glucoside is stored but the mechanism how its integrity as a glucoside is maintained remains unclear (although rigorous experiments were carried out trying to resolve this). *Diabrotica* larvae are able to hydrolyze HDMBOA-Glucoside to release MBOA upon mechanical disturbance (mimicking predation with forceps). Toxic MBOA is also released when *Diabrotica* larvae are actually infected with entomopathogenic nematodes. Both, nematodes as well as symbiontic bacteria (which degrade the beetle larvae and are later fed upon by nematodes), are also capable of hydrolyzing HDMBOA-glucoside and MBOA-glucoside (bacteria only). Sequestered benzoxazinoids reduce nematode infection rates in *Diabrotica* larvae and purified benzoxazinoids applied to nematodes reduce their infectivity. Furthermore, HDMBOA-glucoside and MBOA (but not MBOA-glucoside) decrease survival in nematodes and increase mortality in bacteria. Although not hydrolyzed by the beetle larvae, MBOA-glucoside repels nematodes.

The study is highly integrative and carried out in a very detailed manner which is reflected by a broad array of techniques including biochemical assays (gut extractions), chemical analyses based on LC-MS, and pharmacological as well as behavioral assays involving nematodes and bacteria. The study focusses on interactions mediated by sequestered plant compounds across four trophic levels in probably unprecedented detail. All the experiments were designed very rigorously and elegantly.

Essential revisions:

No major changes are required. Please address all the minor points listed below.

1) To further place this work in the context of the field, consider citing other studies showing that toxic compounds can have effects on the fourth trophic level, such as Harvey et al., 2007, which demonstrates negative effects of dietary nicotine on hyperparasitoids in Manduca sexta, and/or studies showing the activation of activated-defenses such as references in Beran et al., 2014 or Franzl et al. 1989, Experientia.

2) Introduction, third paragraph: remains = remain.

3) Introduction, fourth paragraph: "To the best of our knowledge, the only root feeder known to sequester plant secondary metabolites is the spotted cucumber beetle *Diabrotica undecimpunctata*." -> I don't think this is correct. There is a study by Ali and Agrawal 2017, showing sequestration of cardenolides by cerambycid beetle larvae (*Tetraopes*) from roots of *Asclepias*.

4) In the same paragraph: remove period after *D. undecimpunctata*.

5) Results section, first paragraph: "*D. balteata* and *D. undecimpunctata*, only accumulated trace amounts of BXs..." -> I think the term "trace amounts" is misleading here. According to the figure, *D. undec.* sequestrates about 20% of the benzoxazinoids of *D. virgifera*.

6) In the same section: "Our experiments show that this specialization is associated with the selective accumulation of BXs." -> I think that this statement is too strong as it is only based on three species.

7) Results section, third paragraph: "HDMBOA-Glc release from the exoskeleton did not differ between wild type (WT) B73 and *bx1* mutant fed larvae,..." -> How is this possible? Are low levels of benzoxazinoids still produced in the mutant plants?

8) Figure 1: What are the unlabeled blue datapoints? Additional types of benzoxazinoids?

9) Legend of Figure 2: "BX reactivation by *H. bacteriophora* 24 h after addition of purified metabolites. Inset: MBOA formation in HDMBOA-Glc exposed EPNs." -> there is no inset in Figure 2.

10) Figure 1: the concentration of 2 is much higher than that of 1 in Figure 1 than in Figure 1. This is not explained or discussed at all. Please do so.

11) Regarding Figure 1: did you measure BXs in the frass of the generalist root feeders? This information might be of interest.

12) The legend of Figure 1 and the corresponding text in subsection "Sequestration of maize benzoxazinoids by the western corn rootworm" could be made clearer to indicate precisely when what was measured.

13) Figure 2—figure supplement 1 and subsection "Stabilization of benzoxazinoids in rootworm larvae and reactivation upon attack": the statement that MBOA-Glc cannot be deglycosylated by root-derived hydrolase is a bit too strong since no positive control for any root-derived hydrolase is shown.

14) Second paragraph of subsection "Stabilization of benzoxazinoids in rootworm larvae and reactivation upon attack": Please clarify this experiment, including the data and methods.

15) The paper would benefit from adding a figure that summarizes the findings.

16) Overall, the legends could be made a clearer and include the statistical tests.

---

## [Author Response]

Essential revisions:No major changes are required. Please address all the minor points listed below.1) To further place this work in the context of the field, consider citing other studies showing that toxic compounds can have effects on the fourth trophic level, such as Harvey et al., 2007, which demonstrates negative effects of dietary nicotine on hyperparasitoids in Manduca sexta, and/or studies showing the activation of activated-defenses such as references in Beran et al., 2014 or Franzl et al. 1989, Experientia.

We now added the suggested references.

2) Introduction, third paragraph: remains = remain.

Done.

3) Introduction, fourth paragraph: "To the best of our knowledge, the only root feeder known to sequester plant secondary metabolites is the spotted cucumber beetle Diabrotica undecimpunctata." -> I don't think this is correct. There is a study by Ali and Agrawal 2017, showing sequestration of cardenolides by cerambycid beetle larvae (Tetraopes) from roots of Asclepias.

We thank the reviewers for pointing out the work of Ali and Agrawal 2017, which appeared shortly before submission of our manuscript and therefore escaped our attention. We now include it in the Introduction.

4) In the same paragraph: remove period after D. undecimpunctata.

Done.

5) Results section, first paragraph: "D. balteata and D. undecimpunctata, only accumulated trace amounts of BXs..." -> I think the term "trace amounts" is misleading here. According to the figure, D. undec. sequestrates about 20% of the benzoxazinoids of D. virgifera.

We changed this statement.

6) In the same section: "Our experiments show that this specialization is associated with the selective accumulation of BXs." -> I think that this statement is too strong as it is only based on three species.

Agreed. We have toned down the statement as follows: "Our experiments suggest that this specialization might be associated with the selective accumulation of BXs."

7) Results section, third paragraph: "HDMBOA-Glc release from the exoskeleton did not differ between wild type (WT) B73 and bx1 mutant fed larvae,..." -> How is this possible? Are low levels of benzoxazinoids still produced in the mutant plants?

Yes. BX production relies on the expression of two genes: *bx1* (responsible of about 90% of BX production) and *igl* (10% of BX production). Therefore, *bx1* mutant plants still produce about 10% of BXs compared to WT plants. This is mentioned in the text: "... *bx1* mutant plants which show a 90% reduction in BX levels". The use of the double mutant (BX free) *igl bx1* led to the complete absence of BXs in the larvae and allowed us to rule out a possible production of BXs by the larvae.

8) Figure 1: What are the unlabeled blue datapoints? Additional types of benzoxazinoids?

Yes. We now added this information in the legend of the figure.

9) Legend of Figure 2: "BX reactivation by H. bacteriophora 24 h after addition of purified metabolites. Inset: MBOA formation in HDMBOA-Glc exposed EPNs." -> there is no inset in Figure 2.

Removed.

10) Figure 1: the concentration of 2 is much higher than that of 1 in Figure 1 than in Figure 1. This is not explained or discussed at all. Please do so.

Figure 1 correspond to two different experiments. We did notice some biological variability between experiments, which is likely due to slight differences in rearing conditions or the age of the larvae at the time of larval collection. This is now discussed in the second paragraph of subsection "Sequestration of maize benzoxazinoids by the western corn rootworm".

11) Regarding Figure 1: did you measure BXs in the frass of the generalist root feeders? This information might be of interest.

We did not measure BXs in the frass of the other species. Frass collection is very challenging in *Diabrotica spp.* and requires dedicated method development for the different species together with a large sampling effort. We agree that this information would be interesting to obtain, and will aim to perform such experiments in future studies.

12) The legend of Figure 1 and the corresponding text in subsection "Sequestration of maize benzoxazinoids by the western corn rootworm" could be made clearer to indicate precisely when what was measured.

We reformulated this section and added more details about the procedure in the text and in the figure legend.

13) Figure 2—figure supplement 1 and subsection "Stabilization of benzoxazinoids in rootworm larvae and reactivation upon attack": the statement that MBOA-Glc cannot be deglycosylated by root-derived hydrolase is a bit too strong since no positive control for any root-derived hydrolase is shown.

Done.

14) Second paragraph of subsection "Stabilization of benzoxazinoids in rootworm larvae and reactivation upon attack": Please clarify this experiment, including the data and methods.

This paragraph was edited to improve clarity. More details were added about the procedures.

15) The paper would benefit from adding a figure that summarizes the findings.

Thank you for this comment. We now include a summary figure (Figure 5).

16) Overall, the legends could be made a clearer and include the statistical tests.

Done.